# Building Resilience Through Symphony and Poetry for COVID-19-Encumbered Healthcare Workers: A Taiwanese Qualitative Study

**DOI:** 10.3390/healthcare13091064

**Published:** 2025-05-05

**Authors:** Hui-Yueh Liu, Chun-Kai Fang, Jung-En Peng, Sung-Yuan Cheng, Te-Yu Wu

**Affiliations:** 1Department of Thanatology and Health Counseling, National Taipei University of Nursing and Health Sciences, Taipei 112303, Taiwan; 2Department of Pastoral Care, MacKay Memorial Hospital, Taipei 104217, Taiwan; 3Hospice and Palliative Care Center, Department of Psychiatry, MacKay Memorial Hospital, Taipei 104217, Taiwan; 4Department of Medicine, MacKay Medical College, New Taipei 26245, Taiwan; 5Department of Medical Research, MacKay Memorial Hospital, Taipei 104217, Taiwan; 6Department of Counseling Center, MacKay Memorial Hospital, Taipei 104217, Taiwan; 7Department of Nursing, MacKay Memorial Hospital, Taipei 104217, Taiwan; 8Department of Nursing, MacKay Medical College, New Taipei 26245, Taiwan

**Keywords:** healthcare workers, COVID-19, resilience, symphonic, poetry, qualitative study

## Abstract

**Background:** In 2021, Taiwan’s healthcare workers faced significant stress due to the coronavirus disease 2019 (COVID-19) pandemic. This study aims to enhance healthcare workers’ self-awareness of their stress and improve their work efficiency by using symphonic poetry to inspire resilience. **Methods:** This qualitative study involved in-depth interviews with a panel of physicians, nurses, and specialist nurses. The interviews were conducted using ATLAS—ti 7.5 qualitative analysis software for content analysis. Participants must have attended the “Meeting Mahler, Meeting Myself” concert organized by the medical center. The concert was based on the poetry from the fifth movement of Mahler’s Symphony No. 2, “Resurrection”, and included symphonic poetry and a manual. The manual combined pictures and a complementary interview guide to facilitate participants sharing their pandemic experiences and health. **Results:** A total of 19 healthcare workers participated in the interviews. All participants had attended the institute’s symphonic poetry concert “Meeting Mahler, Meeting Myself”. The average age of the participants was 44.1 ± 11.4 years, and their average working experience was 19.1 ± 13.1 years. The group included nine physicians (47.4%), eight nurses (42.1%), and two specialist nurses (10.5%). The results were categorized into three major themes: “Pandemic-Induced Physical and Mental Exhaustion”, “Symphonic Poetry as a Tool for Building Resilience”, and “Enhancing Mental Toughness to Overcome Adversity”. **Conclusions:** Healthcare workers experienced stress and exhaustion during the pandemic. Symphonic poetry can serve as a supportive tool to inspire resilience and enhance mental toughness among healthcare professionals facing pandemic-related challenges.

## 1. Introduction

The coronavirus disease 2019 (COVID-19) pandemic has significantly altered the working conditions of healthcare workers. Due to the uncertain nature of its pathological mechanisms and high infectivity, healthcare workers often worry about the safety of themselves and their families [1], experience increased workload, and face the emotional burden of patient deaths [2]. These factors have increased psychological symptoms and fatigue, enhancing their willingness to leave high-risk jobs [3]. Additionally, healthcare workers have reported a range of related symptoms, including sleep difficulties, distress, fear, and helplessness [2], as well as high levels of anxiety (46.5%), depression (30.2%), and severe exhaustion (51%) [4]. Furthermore, 7.9% of healthcare workers have experienced post-traumatic stress disorder (PTSD) [5]. As healthcare workers are often at the forefront of responding to these adverse events, their mental health and well-being are seriously affected, which can, in turn, impact their work efficiency. Therefore, their physical and psychological health should be taken seriously [6,7].

The COVID-19 pandemic can be considered a traumatic event, with particularly adverse consequences for the health of healthcare workers. However, researchers indicate that traumatic experiences can lead to positive outcomes and foster resilience and growth. High levels of psychological resilience and effective coping strategies can promote organizational and personal development, reinforcing the importance of well-being [8]. Resilience can manifest in various ways, either positively (growth) or negatively (stress), and is associated with secondary traumatic stress (STS), age, and post-traumatic growth (PTG). In contrast, education and all coping strategies are relevant to STS and PTG [9].

The effects of music reduced stress-related outcomes in two systematic literature reviews and meta-analyses [10,11]. Music intervention significantly reduced physiological (d = 0.380) and psychological (d = 0.349) stress. After moderator analysis, the effect on heart rate (d = 0.456) was more significant than that on blood pressure (d = 0.343) and hormone level (d = 0.349) [10], as well as stress-related outcomes (d = 0.723) [11], indicating that music intervention is meaningful in reducing stress-related outcomes. The term “symphony” originates from the Greek word συμφωνία (symphonia), meaning “harmonious or consistent sound” or “human voice or vocal ensemble”. It is a form of classical music, typically a large orchestral piece consisting of 3–4 movements [12]. The poem “Satz” in the fifth movement of Mahler’s Symphony No. 2, “Resurrection” (Gustav Mahler, 1860–1911: Symphony No. 2 in C minor, “Resurrection”), serves as the thematic core of the creative medium. This hymn, “Die Auferstehung”, was written by the German poet Friedrich Gottlieb Klopstock (1724–1803). Mahler completed the three movements in 1894 and later adapted them into a symphonic poem. The music is rich in poetic and supernatural elements, conveying the literary and melodic motifs of the glorious beauty of victory. It aims to connect people’s emotions and experiences, leading them to a place where they firmly believe in the higher power of heaven to transcend their suffering and demonstrate the triumph and brilliance of life. It expresses a musical realm that breaks through darkness and welcomes light [13]. As a healing method, poetry uses various techniques to promote healing, growth, and transformation. Combining music and symphonic poetry allows healthcare workers to articulate their life experiences and the challenges of contemporary medical practice. It is a therapeutic means to express their unique perspectives and inner strength [14].

This study aims to enhance workplace resilience and spiritual well-being among healthcare workers by using the music and poetry of symphonic poems and picture manuals. These materials will guide healthcare workers in self-awareness and help them explore their inner kinetic energy, thereby improving work efficiency, creating a sense of meaning, and enhancing overall life satisfaction.

Based on the previous statement, the following research questions were established:

How do the experiences of the COVID-19 pandemic impact healthcare workers’ psychological distress in Taiwan? How does using the poetry of symphonic poems help them face the difficulties of the COVID-19 pandemic?

## 2. Materials and Methods

### 2.1. Study Design

This study employs in-depth interviews and content analysis to analyze data until no new categories emerge in the interview content, indicating data saturation.

### 2.2. Participants

This study uses convenience sampling to recruit healthcare workers (physicians, nurses, and specialist nurses) from a medical center. The inclusion criteria are as follows: (1) full-time clinical medical staff, (2) who have worked at this institution for at least one year, (3) have participated in the “Meeting Mahler, Meeting Myself” concert of Mahler’s Symphony No. 2 “Resurrection” organized by the researchers’ institute (the course lasted sixty minutes; the researchers guided the participants to listen to and meditate on Mahler’s symphony with a poetry manual for thirty minutes; afterward, the participants quietly reflected on their physical and mental conditions during the COVID-19 pandemic and wrote down their feelings on small cards for thirty minutes; there were 140 participants, of which 74 were in the experimental group and 66 were in the control group), (4) can understand written and spoken Chinese, and (5) agree to participate in this study. The exclusion criteria are as follows: those who do not agree to participate in this study. After the employees agree to participate, the researchers will explain the study process and obtain their signed consent form. Willing employees will be interviewed using a snowball sampling method.

### 2.3. Study Material

This study uses the poetry from the fifth movement, “Satz”, of Mahler’s Symphony No. 2 as the creative medium. The researchers employ music listening, analysis, symphonic poetry, and guiding sentences. Based on Frankl (2014), the master of existential meaning, “the individual is the only one to decide about the meaning of their life and takes responsibility for creating and choosing their unique meaning. The ability to determine the meaning of a situation can make a positive outcome from the worst of situations” [15]. Therefore, we discussed the existential meaning guidelines as the core of the interview guidance (e.g., “How did you feel about your medical work in the pandemic? How has facing the pandemic been in psychological distress? Has it brought any changes to you? How does using the poetry of symphonic poems help you face the difficulties of the COVID-19 pandemic?”) to guide participants. The guiding words and visual aids were compiled into PowerPoint (PPT) slides and a manual to present the concepts of Mahler’s symphony, transitioning from darkness (tired, frustrated, etc.) to dawn (now, warm, radiant, etc.) [16]. The participants were divided into seven groups when they listened to the fifth movement, “Satz”, of Mahler’s Symphony No. 2 in a comfortable lying position in a room with 3–5 people. For epidemic prevention, participants had to maintain an indoor space of at least 1.5 m per person. Wearing a mask was not mandatory, but participants with relevant cold symptoms or who could not maintain social distance from unspecified people did have to wear one [17]. The researchers softly narrated according to the dynamics of the symphony and the PPT presentation. The guiding words facilitate a quiet and in-depth self-dialogue, allowing participants to gradually express their feelings about the current medical situation without stress. Additionally, this study does not use artificial intelligence to assist in generating interview content or processing data for participants.

### 2.4. Data Collection Process

Recruitment for this study was reviewed and approved by the Medical Center Institutional Review Board (21MMHIS312e). The “Meeting Mahler, Meeting Myself” concert was held six times from April to May 2022. Participants attended the concert within one week, and the project leader explained the researchers’ motivation, purpose, and procedures. After participants signed the consent form, the researchers scheduled an interview time and arranged a quiet and private environment. The project leader conducted a literature review and developed an interview outline: (1) How does participating in the “Meeting Mahler” project impact your impression or understanding of resilience at work? (2) How does resilience influence past distress or difficulties at work? (3) How did participating in “Meeting Mahler” change your psychological state? (4) How does the meaning of the poem “Resurrection” in the fifth movement of Mahler’s Symphony No. 2 affect your current mindset? Each scheduled interview lasted 30 min and used the researchers’ materials to guide participants in exploring the impact of COVID-19 on their work. With the prior consent of the participants, the researchers recorded the entire interview and documented meaningful nonverbal responses. After each interview, the researchers listened to the whole recording and transcribed the interview verbatim. The verbatim transcript was provided to each participant for confirmation. Subsequently, two researchers used ATLAS Ti 7.5 qualitative analysis software to identify meaningful content, collect codes, and analyze data.

### 2.5. Rigor

The information production was based on the belief that “The individual is the only one to decide about the meaning of their life and takes responsibility for creating and choosing their unique meaning” [15]. Therefore, the researchers guided participants in articulating the meaning of their healthcare work during the COVID-19 pandemic.

The study’s rigor was assessed based on Lincoln and Guba’s four criteria for trustworthiness [18]. In terms of credibility, one of the researchers holds a master’s degree in art, has a foundation in symphony conducting research and training, and possesses professional quality in analyzing or interpreting classical music works. She performed Mahler’s Symphony No. 2, “Resurrection”, at the Taiwan National Concert Hall in 2021, focusing on musical tension, instrumental arrangement, and poetic expression. Additionally, she uses music and growth activities to guide medical students’ self-awareness and enhance their spiritual exploration experience. Her role as a hospital clergyman also facilitated a quick establishment of trust with the participants, increasing their willingness to share their experiences. The other researcher is a university professor with expertise in qualitative interviewing and data analysis, which further increased the trust of the interviewees in the researchers, allowing participants to express their inner feelings and enhancing the credibility of the data. Regarding transferability, this study recruited participants from various medical center departments, increasing the breadth and applicability of the data. Additionally, we recorded the entire interview process and wrote a transcript. ATLAS Ti 7.5 qualitative analysis software was used to ensure the reliability of the data collection and analysis process. We also maintained a record of decisions made during the study for dependability. The researchers and two qualitative research experts continuously confirmed and reconfirmed the data and results, ensuring that the participants’ experiences with COVID-19 in their work and lives were fully expressed. Simultaneously, the coding results were mailed to the participants, who were reminded to reply to the research team if they had any questions. Finally, the researchers discussed and confirmed the codes, categories, and results with another researcher, a psychiatrist, to enhance the confirmability of this study. Therefore, it also conforms to the trustworthiness factors in qualitative research [19], which are emphasized by authors working together to establish trustworthiness.

### 2.6. Ethical Consideration

The Research Ethics Committee of the Medical Center Institutional Review Board approved this study. The researchers explained the study’s purpose and individual rights and obtained written consent from all participants. Participants were informed that the information obtained during the interviews would be kept confidential and secure. They had the right to refuse to answer questions about sensitive issues. Additionally, if participants experienced emotional discomfort during the interview, the researchers would immediately stop and arrange another time to revisit; otherwise, participants could withdraw from the study.

## 3. Results

A total of 19 healthcare workers participated in this interview. All participants attended our institute’s symphonic poetry concert “Meeting Mahler, Meeting Myself”. Five of the participants were male (26.3%), and fourteen were female (73.7%). The average age was 44.1 ± 11.4 years, and the average working experience was 19.1 ± 13.1 years. The participants included nine physicians (47.4%), eight nurses (42.1%), and two specialist nurses (10.5%) (Table 1). Using ATLAS ti 7.5 qualitative analysis software, the researchers’ results were summarized into three major categories: “The pandemic aggravates physical and mental exhaustion”, “Symphonic and poetry can build resilience to stress”, and “Enhancing mental toughness to overcome adversity”. The following sections explain the connotation of each category in detail (Table 2).

### 3.1. First Category: The Pandemic Aggravates Physical and Mental Exhaustion

Mahler’s original text states “Mit Flügeln, die ich mir errungen Werde ich entschweben. Sterben werd ich, um zu leben!” because the poem contains the faith and hope of resurrection. Therefore, in the guiding symphony poem by researchers, we use the guiding words to lead the participants into the hope and enthusiasm of resurrection. The guiding symphony poem includes the words: “In this world of transcendence of the spirit and pandemic stress, let the symphony and poem lead you into feeling everything, a conversation with yourself…” Due to the increased psychological stress caused by the COVID-19 pandemic, medical decisions are frequently adjusted, making the work of healthcare workers uncertain and anxiety-inducing. Amidst the tedious medical affairs, busy clinical work is often filled with frustration and helplessness.

#### 3.1.1. The Pandemic Increases the Burden of Medical Work

During the pandemic, healthcare workers had to adapt to policy changes, including the uncertainty of the effectiveness of protective equipment and the risk of exposure to the virus, as well as adjustments to departmental work and restrictions on family visitors. These factors require significant time and physical effort to manage patient care, placing an even more substantial burden on healthcare workers. The following are statements from healthcare workers:*Facing the pandemic, the work is cumbersome, and the stress!* (Nurse 6).*Caring for patients regardless of specialty.* (Nurse 7).*Many people are unaware and afraid of the disease.* (Nurse 3).*Colleagues are diagnosed one after another. We are also very strained in the nursing schedule* (Nurse 5).

#### 3.1.2. Dual Stress at Work and Home Deepens Exhaustion

Healthcare workers typically bear a heavy workload and face frequent policy changes and evaluation requirements. They struggle to balance family and work, which exacerbates their physical and mental exhaustion. The following are their perspectives:*Whether it’s about colleagues or policies… I feel very powerless…*(Nurse 7).*I encounter setbacks every day. How do we face these setbacks?* (Nurse 8).*I work so hard to help and take care of patients, but I cannot take care of my family…* (Nurse 2).

### 3.2. Second Category: Symphonic and Poetry Can Build Resilience to Stress

The guiding symphony poem includes the lines: “Now, are you still immersed in fatigue and boredom? Or has it made your life stagnant? Or is there any transformative power? Alternatively, has the love you first devoted to the medical field allowed you to regain your strength and keep moving forward?” Healthcare workers can discuss how symphonic poetry helps them adjust and release stress in the face of the pandemic and work tension.

#### 3.2.1. Combination of Symphony and Poetry on Stress Reduction and Healing

Mahler’s symphony features dynamic fluctuations, with a chorus of music accompanied by guiding words and images that transition from darkness (tiredness, frustration, etc.) to dawn (warmth, brilliance, etc.). This allows participants to experience the changes and imagery of personal situations, recognize the uncertainty of life’s journey, and feel a release of mental stress. Participants shared their thoughts:*When the pictures and words are paired with the symphony… I think there is a sense of being healed…* (Physician 2).*When I listened to the symphony and read the poetry in the manual, I felt that life is like a journey with peaks and valleys…* (Physician 5).*Poetry annotations accompany Mahler’s music and symphony… feel the changes and imagery the symphony aims to convey. This is the power of music…* (Physician 9).

#### 3.2.2. The Tension of the Symphonies Encourages Me to See Hope from the Trough

The symphony’s melody has a choral fugue style, using the overlapping of choir parts and the loud timbre of brass instruments to provide an exhilarating auditory experience. The energy of the symphony helps healthcare workers experience the therapeutic effects of music, which can transform darkness into light. Healthcare workers shared their thoughts:*Mahler’s symphonies and poems… What I find most striking about this piece of music is the fugue. The fugue in music creates a sense of breaking through the sky…* (Nurse 7).*“This symphony encourages me and can transform these setbacks or stagnations into positive energy and acceptance of differences…* (Nurse 1).

#### 3.2.3. Symphonic and Poetic Guidance and Encouragement Can Overcome Setbacks

When guiding the symphony poem, “The haze is about to pass, all fatigue, frustration, and hardship will be buried; the soul will awaken again, and the fire of enthusiasm will reignite”, the researchers lead participants back to reality. The symphony poem guides the sense of the meaning of suffering, allowing participants to feel the significance of life. Through the experience of real suffering, they can once again realize their self-worth and the meaning of life, leading to beautiful actions. Participants described the impact of symphonic poetry on them:*Mahler transformed the suffering of life into the power to overcome it through his symphony… which is to seek God to transcend personal suffering.* (Physician 6).*“In all difficulties or moments, there is a feeling that everything will be alright. It means that after being in darkness for a long time, light will eventually appear. This includes frustration in medical work… As long as you keep moving forward, there will always be growth.* (Physician 9).

#### 3.2.4. Feeling the Power of Transformation, Like a Voice from on High

Continuous dedication to medical work can be seen as planting hope for the future. When these setbacks are overcome, one will experience a renewed sense of self-belief and discover that it is not lost but has transformed into a new beginning. Participants shared their ideas:*Even if he encounters an unsatisfactory situation, as long as he continues to work hard, he will achieve a bright outcome, much like the symphony…* (Physician 7).*I feel exhausted… The fluctuations in the notes and the composition gradually repair my heart… I think some of my strength has returned…* (Nurse 3).*The theme of the symphony is resurrection. In the end, we will see the light. Whether in life or medical work, everything can be resolved…* (Nurse 4).

### 3.3. Third Category: Enhancing Mental Toughness to Overcome Adversity

Through the beautiful melody of the symphony, the “hearts” of healthcare workers can temporarily find solace, become aware of their own psychological emotions, and briefly focus on their inner emotional anxiety or worry. The impact of emotions is continuous. When the symphony resonates with their personal experiences, their thoughts and decisions are altered, leading them to reinterpret and integrate into their environment and life stories.

#### 3.3.1. “Change” Is Gathering Positive Energy

While listening to the symphony, healthcare workers were encouraged to “believe” that not all is lost but that this is a moment, and we can get through it together. In the continuous dedication to medical work, hope for the future can be planted. When these setbacks are buried, believing in oneself is not lost but allowed to transform into a new beginning gradually. Statements from healthcare workers describing the impact of the symphony on their psychology:*We will go through a dark period… Just like the soprano sang in the third movement, ‘Believe! My heart! You must believe you have not lost everything… but about transforming these challenges into something better, you can see the eternal light… *(Nurse 7).*After hearing the symphony, I feel calmer and more released. I feel like I have more energy and can face new challenges.* (Physician 2).*When you encounter setbacks… You need help in all aspects… otherwise, you cannot extricate yourself.* (Nurse 8).

#### 3.3.2. The Power of the Resurrection Was to Rekindle Hope

The healthcare workers described that while listening to the symphony, they felt freed from the haze, rekindled, inspired, and empowered. The symphony enters the theme, and the beautiful melody brings out the hope and power of life. Take a quiet moment to listen to the music and experience your mood through the guiding words. Although the pandemic is full of challenging medical situations, the persistence of life is so beautiful and meaningful. Statements from participants:*Through this symphony, I can elevate myself or transcend my suffering…* (Physician 6).*I like the resurrection theme in the fourth movement, ‘You will be resurrected, my heart is in a moment, and everything you strive for will lead you to see God’. … the resurrection of Jesus: rebuilding ourselves again ignites hope for the future… God has been guiding us from the original intention in medicine.* (Physician 2).*I prefer the part at the end of the symphony… the problematic situation will pass… You will be regenerated, so that part is quite encouraging.* (Nurse 6).*It is a compelling, sonorous melody and sound, so you will not always stay in that trough… Let yourself have the strength to move forward.* (Specialist Nurse 2).

#### 3.3.3. Assuming Medical Responsibility Under Stress

In the guide of the symphony poem from the manual, it reads “Believing that you are in this medical environment is not in vain! We ignite and encourage each other; every day of our lives is glorious and beautiful”. Through the beautiful melody of the music, the “hearts” of medical staff can take a temporary rest, be aware of their own psychological emotions, and briefly pay attention to their inner emotional anxiety or worry. The impact of emotions is continuous. When music meets them at this moment, their thoughts and decisions change accordingly. Changes affect the re-interpretation and integration of the environment and one’s life story.

The healthcare workers recalled the past when they first entered this sacred medical work. They persisted in their positions because of their initial love. They did not waver or panic. All because of their initial persistence, they got to where they are now and responded to the divine call again, rekindling a new journey in the challenging pandemic. The healthcare workers stated the following:*I felt that the patient’s life and death are closely connected with us. Everything we do should be regarded as very solemn and very sacred.* (Physician 3).*From the symphony… Let’s be capable and believe we can take on some work responsibilities.* (Nurse 6).*In Mahler’s symphony, you can feel the bird’s wings waving. It is not just one bird but a flock… a medical team is essential in medicine. This is teamwork… *(Physician 6).

## 4. Discussion

This study uses the theme of the fifth movement, “Satz”, of Mahler’s Symphony No. 2. The researchers guide the 19 participants, healthcare workers at the medical center, to gradually share their feelings about COVID-19 and their current work situation without any stress through music appreciation, music analysis, symphonic poetry, and words of guidance. The researchers’ results are summarized into three themes: “The pandemic aggravates physical and mental exhaustion”, “Symphonic and poetry can build resilience to stress”, and “Enhancing mental toughness to overcome adversity”.

### 4.1. First Category: The Pandemic Aggravates Physical and Mental Exhaustion

The first theme of this study is “The pandemic aggravates physical and mental exhaustion”. Since the pandemic has altered the clinical norm, the uncertainty of work and pandemic risks have increased the work burden of healthcare workers. The stress at work and home has also made healthcare workers feel physically and mentally exhausted. Studies have shown that healthcare workers are worried about the risk of infecting family and friends [20]. Additionally, 17.8% of healthcare professionals required psychological or psychiatric support and requested psychological help. Those who required psychological support had more negative attitudes and perceptions towards their work, even experiencing “burnout” [21]. They are also concerned about the fear of stigmatization and work-related risks, which can lead to feelings of isolation and social deprivation [20]. Another study demonstrated that the lower the degree of contact with remote patients and the shorter the time, the higher the levels of vigor, organizational support, and trust in equipment and infection control initiatives; it can be predicted that medical staff caring for SARS (severe acute respiratory syndrome) will exhibit lower avoidance behavior, emotional exhaustion, and state anger [22].

Furthermore, researchers indicate that during the COVID-19 pandemic, higher trust in communication regarding the availability of personal protective equipment (PPE) and infection prevention strategies, including information transparency and clarity, uniformity, and consistency of medical care guidelines, and the provision of face-to-face teaching via digital resources, along with appreciation and support for the work performed among healthcare workers, can reduce emotional exhaustion during the pandemic [23]. The findings of this study align with these results. Therefore, during future contagious disease pandemics, policy formulation and actions concerning equipment, mental health, and safety awareness of medical personnel can mitigate the threat of infectious diseases to healthcare workers and prevent physical and psychological exhaustion.

### 4.2. Second Category: Symphonic and Poetry Can Build Resilience to Stress

Through the guidance of symphonies and poetry, participants can feel the tension and continuity of the symphony’s melody. Music allows them to experience the power to transform stress, face the fear caused by the pandemic, and inspire the recovery power of healthcare workers to continue their medical work. Many believe music and poetry can heal a person, as soothing words can comfort and ease traumatic experiences. When needed, poetry can naturally emerge and move people, allowing them to describe the fantastic changes they experience [14]. Mahler’s fifth movement, “Satz”, is a choral symphony that combines orchestra, choir, soloists, and chorus. There is hope and comfort in the poetry of the Resurrection Symphony No. 2; the first movement (Allegro maestoso) opens with solemn low strings and builds up a sense of motion; the second movement (Andante moderato) brings a graceful, dance-like theme, and successively more restless themes. The symphony ends with the same ecstatic and glorious mood for the chorus and orchestra, bringing listeners’ emotional shock from low to high [24]. This music showcases the composer’s creative techniques with romantic melody lines and instrumental arrangements that expand the range of enhancement from one voice to a four-part fugue harmony. The cyclical flow of music brings out eternal and endless energy, allowing the listener to feel the transcendence of the musical kinetic energy [16,24,25].

### 4.3. Third Category: Enhancing Mental Toughness to Overcome Adversity

Healthcare workers are susceptible to the psychological impacts of the health crisis during the COVID-19 pandemic, and resilience is a potential protective factor in coping with psychological distress during crises. Factors that enhance their resilience include “Professional Training”, “Support and Wellbeing Measures”, “Reorganization of Services”, and “Professional Acknowledgment” [26]. While listening to the symphony, healthcare workers were encouraged to “believe” that they had not lost everything. This is a short moment of experiencing the pandemic; they can go through it together. The resilience of healthcare workers from infectious diseases over the years mainly stems from their moral purpose and duty, connections with peers, family, community, and media, emotional support in the organizational culture, and experiential learning. This time shall pass, and there is potential for growth through heartfelt reflections [27]. Factors influencing physicians’ passion during infectious disease pandemics include meaningful work, work–life balance, challenging tasks, task accomplishment, a positive environment, psychological support, and spiritual drive [28]. Additionally, research on the psychological experiences and reactions of physicians who have been infected with COVID-19 showed that among the “adaptive emotional reactions” is feeling happy to have a medical job and gaining resilience because, in times of crisis, medical personnel, although infected, still maintain hope of recovery and are very happy that their medical work can help humanity. In “adaptive spiritual reactions”, some physicians find a vital role through “attachment to God”, “recourse to religious beliefs”, and “trust” through constant prayers, knowing that through “trust” in Him, everything will get better [29]. These results are consistent with this study. The resilience of healthcare workers in the past mainly stems from their professional identity, collegial support, effective communication from supportive leaders, and self-flexibility to engage in self-care and experience growth. These insights can provide medical managers with strategies to develop physicians’ enthusiasm for work, thereby improving the quality of the medical environment.

### 4.4. Limitations and Recommendations for Future Researchers

The present study has several limitations. First, since most of classical music’s creative and poetic meanings come from the history of Western music or the self-interpretation of music performers, few published academic articles can be used as references for this study. Second, since this study used Mahler’s symphony poems to guide participants, it may have potential implications for participants’ thoughts. If this research method is to be used in the future, a more rigorous design may be required. Third, the results of this study can only reflect the opinions of participants and cannot necessarily represent the opinions of healthcare workers in other regions. However, the results of this study suggest that more diversified intervention measures will be needed in the future to support medical workers’ resilience and spiritual care. Classical art can also be heard in concert halls and used in medical work.

## 5. Conclusions

The participants in this study, who have experienced the stress and helplessness associated with the COVID-19 pandemic, found that through the guidance of symphonies, poems, and picture manuals, they felt a sense of divine inspiration and strength derived from the symphony melodies. Listening to symphonies and poetry helped them transform stress and confront the fears induced by the pandemic. Psychological resilience prevented and reduced the negative psychological impact during the pandemic and enhanced their ability to continue their medical work. Furthermore, this intervention promoted positive personal growth and equipped them with the skills to cope with future unknown diseases. This intervention can help clinical healthcare workers improve their adaptability to infectious diseases and be a valuable reference for medical strategic planners. Currently, the hospital involved in the study is a Christian hospital; the hospital’s pastoral care workers participate in a nursing fellowship, a physician fellowship, a specialist group, and 14 other online revival prayer groups (RPGs) for physicians. They meet twice a month for three to four people in the RPG. The hospital’s doctors, nurses, and nursing specialists are connected through these groups. They gather to sing a hymn, pray, and connect spiritually to enhance workplace resilience and spiritual care. The COVID-19 pandemic has further suggested the effectiveness and strategy of regularly implementing prayer meetings in these hospitals.

## Figures and Tables

**Table 1 healthcare-13-01064-t001:** Demographic characteristics of the participants (*n* = 19).

Code	Department	Age (Years Old)	WE (Years)
P1	Traditional Chinese Medicine	33	4
P 2	Traditional Chinese Medicine	33	3
P 3	Oncology	40	15
P 4	Critical care	62	35
P 5	Otolaryngology	59	34
P 6	Obstetrics and Gynecology	39	10
P 7	Cardiovascular surgery	55	22
P 8	Surgery	38	6
P 9	Surgery	30	3
N1	Outpatient clinic	58	38
N2	Otolaryngology	46	23
N3	Chest ward	28	4
N4	Hemodialysis department	56	36
N5	Pediatric ward	45	24
N6	Cardiovascular ward	49	26
N7	Outpatient clinic	38	15
N8	Nursing department	58	40
SN1	Urology	27	5
SN2	Administration manager	44	19
Average		44.1 ± 11.4	19.1 ± 13.1

PS: P = physician; N = nurse; SN = specialist nurses; WE = working experience.

**Table 2 healthcare-13-01064-t002:** Categories and subcategories based on content analysis.

Quotations	Sub-Category	Category
*Facing the pandemic… cumbersome… stress!* (Nurse 6) *regardless of specialty.* (Nurse 7) *unaware and afraid of the disease.* (Nurse 3) *diagnosed one after another… very strained in the nursing schedule* (Nurse 5)	The pandemic increases the burden of medical work.	The pandemic aggravates physical and mental exhaustion.
*I feel very powerless…* (Nurse 7) *setbacks every day. How do we face these setbacks?* (Nurse 8) *hard to help and take care of patients, but I cannot take care of my family…* (Nurse 2)	Dual stress at work and home deepens exhaustion.	
*pictures and words are paired with the symphony… healed…* (Physician 2) *the symphony and poetry…; …a journey with peaks and valleys… *(Physician 5) *Poetry accompanies Mahler’s poetry and symphony… feel the changes and imagery… the power of music…* (Physician 9)	Combination of symphony and poetry on stress reduction and healing.	Symphonic and poetry can build resilience to stress.
*Mahler’s symphonies and poems… The fugue in music… breaking through the sky…* (Nurse 7) *encourages me and transform these setbacks or stagnations into positive energy…* (Nurse 1)	The tension of the symphonies encourages me to see hope from the trough.	
*Mahler transformed the suffering of life into the power to overcome it through his symphony… which is to seek God to transcend personal suffering.* (Physician 6) *everything will be alright… This includes frustration in medical work… *(Physician 9)	Symphonies and poetry can inspire me to overcome setbacks.	
*unsatisfactory situation… we will achieve a bright…, much like the symphony…* (Physician 7) *I feel exhausted… repair my heart… some of my strength has returned…* (Nurse 3) *the symphony is resurrection… we will see the light. Whether in life or medical work, everything can be resolved…* (Nurse 4)	Feeling the power of transformation, like a voice from on high.	
*a dark period… Believe! …transforming these challenges… *(Nurse 7) *symphony,…more energy… new challenges.* (Physician 2) *setbacks… You need help in all aspects…* (Nurse 8)	“Change” is gathering positive energy.	Enhancing mental toughness to overcome adversity.
*You will be resurrected…* (Physician 2) *symphony, … transcend my suffering…* (Physician 6) *You will be resurrected… *(Physician 2) *You will be regenerated…* (Nurse 6) *compelling, sonorous melody and sound… strength to move forward.* (Specialist Nurse 2)	The power of the resurrection was to rekindle hope.	
*Everything… solemn and sacred. *(Physician 3) *Let’s be capable and believe… take responsibility. *(Nurse 6) *In Mahler’s symphony, … This is teamwork…* (Physician 6)	Assume medical responsibility under stress.	

## Data Availability

The data presented in this study are available at the request of the corresponding author due to restrictions imposed by the scientific ethics committee.

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
