# Peer review of "Building Resilience Through Symphony and Poetry for COVID-19-Encumbered Healthcare Workers: A Taiwanese Qualitative Study"

_healthcare, 2025, doi:10.3390/healthcare13091064_

Round 1
Reviewer 1 Report
Comments and Suggestions for Authors
Thank you for your interesting work! It is an original project which is written clear and sound.
I have just a few comments.
In line 110 you mention 'PPT slides'. I assume you mean powerpoint, but it is not clear. Please use the full term before you use an abbreviation.
It is not clear what the time was between the concert and the interviews. Please add that to the text.
Can you please ellaborate more about the meaning of this intervention. I assume it can be much broader than infectious disease. How and in which situation could this be a helpful intervention? And for who? What is the overall meaning of this intervention in the field of health care?
Do you have any suggestions for follow-up of the project or research concerning this theme?
Reviewer 2 Report
Comments and Suggestions for Authors
Thank you for submitting this paper for publication consideration. The topic is interesting and important.
Abstract
- Suggest including SD when means are provided
- Suggest avoiding "significant" under conclusions as you did not test for that. I agree that it was significant, but this is a qualitative paper and I don't want readers to be confused.
Literature review
- Line 71- is "song" the best word? Perhaps piece?
- Line 82 - "This study aims" - suggest making a purpose statement and research question. These should be explicitly stated
Method
- 3-5 people per group - but how many groups? How many people did the researchers offer to attend workshops?
- Perhaps include additional information about the intervention itself. In current format, the protocol cannot be replicated.
- How long were the sessions?
- Researcher or researchers? Please check for consistency.
- Did an expert in qualitative research look over the interview questions? How were these questions determined?
- What language were interviews conducted in? Were interviews translated/interpreted to English? If so, who did these and were there checks?
- Rigor section - please describe the positionalities of the researchers as they are interpreting the data. Their positionalities are important and that needs to be explicitly stated for readers (I might identify as Male, able-bodied, European decent, etc.)
- Because of the pandemic, were there any protective measures taken? Masks? If so, how might these have impacted the experience and results?
- How did the authors work together to establish trustworthiness?
- Did participants review the findings? (They reviewed the transcripts, which is good - but having them review the results would strengthen the paper.)
Results
- Results are organized in a clear and coherent manner
- Were there any negative findings?
Discussion
- Line 379 - suggest deleting "voice of God" as this is subjective and may be centered in the participants' worldview
- Another limitation is a lack of negative findings. Perhaps participants did not feel comfortable sharing their critical experiences?
Thank you for your work. I hope to read more from the authors in the future and wish them well.
Reviewer 3 Report
Comments and Suggestions for Authors
Thank you for the opportunity to read and review this manuscript. Overall the manuscript is well written. The research topic is interesting and innovative. However, some major revisions are needed before I can recommend this manuscript for publication. Following are the comments that I hope would be useful to the authors.
Introduction:
- This section mainly focused on the potential influence of the COVID-19 pandemic as a traumatic event on healthcare workers’ psychological health, and background information of the symphony and poetry used in this study. Although I find the content relevant and important, I think it would also be helpful to see a brief review of the status quo of research on the application of music and poetry in psychological therapy and interventions, the advantage of these approaches and why more investigations such as the present study are necessary.
- p.2 – In the last paragraph of the introduction the authors stated that ‘this study aims to enhance workplace resilience and spiritual well-being among healthcare workers by using…’. To me this is more like the aim of developing an intervention rather than conducting the research. I would suggest a more accurate study aim indicating what will be investigated and specific objectives which can further justify the study design (e.g., the interview outline) and data analysis.
Materials and Methods
- p.3 – In section 2.3 the authors described the application of guiding sentences and words. This appears to be an important and novel element of the study materials. It would be helpful to see more details on how this part of the study materials was developed (e.g., the theoretical or empirical basis).
Results:
- p.6, line 192–196: The guiding symphony poem explicitly led the participants to pay attention to certain feelings such as fatigue, heaviness, boredom and frustration, and asked if their mood has stagnated in a medical situation filled with exhaustion. My concern is that this might result in biased responses of the participants during the interview, which is similar to the effects of leading or suggestive questions in interview studies (see Cairns-Lee, Lawley, & Tosey, 2022; Erčulj & Šulc, 2025). This also applies to other quotes of the guiding symphony poem (e.g., p. 7, line 224-227). Leading questions in interview studies can seriously undermine the validity of study results and conclusions. I would encourage the authors to further justify the usage of these leading or suggestive statements/questions and discuss their potential effects on the study results.
Discussion
- p.12, section 4.4 – I would suggest a more comprehensive discussion on the potential weakness of study design and methodology and how these limitations may impact the study findings. More specific recommendations for future research based on the findings of the present study are also encouraged.
Round 2
Reviewer 2 Report
Comments and Suggestions for Authors
The authors' positionalities are important and that needs to be
explicitly stated for readers (I might identify as Male,
able-bodied, European decent, etc.)
Reviewer 3 Report
Comments and Suggestions for Authors
I would like to thank the authors for their efforts in revising this manuscript. As far as I can see the manuscript has been greatly improved and I’m glad to recommend it for publication in the current form. Only a minor point:
p.2, line 65 – 66: ‘Two systematic literature reviews…stress-related outcomes.’ Please double check the grammar of this sentence.
